# Egg consumption and bone mass density among the elderly: A scoping review

**Mobolaji T. Olagunju**[1], **Olunike R. Abodunrin**[1], **Ifeoluwa O. Omotoso**[2], **Ifeoluwa E. Adewole**[3], **Oluwabukola M. Ola**[3], **Chukwuemeka Abel**[3], **Folahanmi T. Akinsolu**[3,4] *

**1** Nanjing Medical University, Nanjing, China, **2** Chongqing Medical University, Chongqing, China, **3** Lead City University, Ibadan, Nigeria, **4** Nigerian Institute of Medical Research, Lagos, Nigeria

☯ These authors contributed equally to this work.
‡ These authors also contributed equally to this work
* Folahanmi.tomiwa@gmail.com

**Data Availability Statement:** Available in the manuscript

**Funding:** The authors received no specific funding for this work.

## Abstract

Eggs offer a range of essential nutrients that could support skeletal health as individuals age. Maintaining bone density is crucial for reducing the risk of fractures and improving overall mobility and quality of life in later years. Understanding the potential benefits of habitual egg consumption on bone mass density among older people is essential, given that the natural decline in bone mass density occurs with age. This area of research has not garnered sufficient attention basically because of the mixed reactions and conflicting reports about the safety of egg consumption especially among the older adults. This scoping review aims to systematically examine the existing literature to map the evidence regarding the association between habitual egg consumption and bone mass density in older adults' individuals. The scoping review adhered to the Preferred Reporting Items for Systematic Reviews and Meta-Analyses Extension for Scoping Reviews (PRISMA-ScR) guidelines to ensure methodological rigor and transparency. Five electronic databases were searched for published pieces of literature. While high egg intake has been linked to increased mortality and dyslipidemia, eggs contain compounds like Ovo transferrin and carotenoids that may benefit bone health. As aging increases vulnerability to bone fragility and fractures, it's crucial to provide comprehensive dietary recommendations. The complex relationship between egg consumption, cholesterol, and health highlights the need for nuanced assessment. Overall, eggs present a potentially valuable dietary component for promoting bone health in aging populations. Limited research on the link between egg consumption and bone mass density in older adults highlights the need for further investigation. Concerns about cholesterol have overshadowed potential benefits. Given aging populations and bone health challenges, exploring eggs' role in preventing falls and fractures is essential for a proactive approach to older adults' well-being.

**Competing interests:** The authors have declared that no competing interests exist.

## Introduction

Eggs are often considered nutritional powerhouses, and their potential contribution to bone health is increasingly recognized [1], attributed to the rich array of essential nutrients found in eggs that play pivotal roles in improving bone strength and overall skeletal well-being [2].

Firstly, eggs contain Vitamin D, a vital nutrient known for its role in calcium absorption. Calcium, in turn, is a cornerstone in bone mineralization and density [3]. Adequate Vitamin D intake ensures that the body's calcium is effectively absorbed and utilized in bone-building, thus promoting bone health [4]. Furthermore, eggs are a source of Zinc, a mineral crucial in supporting bone formation and repair processes [5]. Zinc contributes to synthesizing collagen, a protein essential for maintaining bone structure and integrity. Its involvement in bone health has been highlighted in various studies [5].

Eggs also feature osteogenic bioactive components such as lutein and zeaxanthin. While these compounds are better known for their role in eye health and reducing the risk of age-related macular degeneration, their potential impact on bone health is emerging. By mitigating oxidative stress and inflammation, lutein and zeaxanthin may indirectly support overall health, including bone health [6,7].

It's worth noting that previous research primarily focused on individual nutrient components of eggs, such as calcium [8], protein [9], and vitamin D [10] about bone health. These nutrients are critical for maintaining strong bones, and their presence in eggs underscores the potential benefits of egg consumption [3,4,8–10]. Historically, whole eggs have faced scrutiny due to their high cholesterol content, and concerns about their impact on cardiovascular health have been the subject of research [11–14]. However, recent studies have challenged this notion, suggesting that moderate egg consumption may not significantly increase the risk of heart disease [13,14]. Yet, the specific associations between habitual egg consumption and bone health remain relatively uncharted in nutrition research.

Peak bone mass, reached in the third or fourth decade of life [7,15], is a critical foundation for skeletal health. However, as individuals age beyond this point, a consistent and natural decline in bone mass density impacts both women and men [15,16]. This age-related reduction in bone mass density plays a pivotal role in the increased susceptibility to vertebral fractures among the older adults' population [34]. Understanding the factors that influence bone health during aging is paramount, and dietary habits are a contributing factor. Habitual egg consumption has emerged as a potential.

Dietary strategy to promote and maintain good bone health, particularly among older people [17]. The unique nutritional composition of eggs, including vital nutrients like Vitamin D, Zinc, and protein, may play a significant role in supporting bone density and minimizing age-related bone loss [18].

This scoping review aims to systematically examine the existing literature to map the evidence regarding the association between habitual egg consumption and bone mass density in older adults' individuals. By synthesizing the available research, we can gain valuable insights into the potential benefits of including eggs as a dietary component in promoting skeletal health during aging.

## Methodology

The scoping review adhered to the Preferred Reporting Items for Systematic Reviews and Meta-Analyses Extension for Scoping Reviews (PRISMA-ScR) guidelines [19] to ensure methodological rigor and transparency.

### Research question

The following inquiry steered the review:

What evidence exists regarding the association between egg consumption and bone mass density in older individuals?

### Articles identification

The initial search was conducted in June 2023 on five electronic databases: Scopus, CINAHL, Web of Science, Medline, and PubMed. The search was performed using the following search strategies enumerated in Appendix (See S1 Appendix). No protocol was published for this review.

### Eligibility criteria and article selection

The literature obtained through database searches was imported into Rayyan's reference management software. Duplicates were removed using the "duplicate items" function. Three independent reviewers (OA, IO, and MT) conducted title and abstract screening, following the eligibility criteria set for this review. A full-text review of the remaining publications was then completed independently by five researchers (OA, IO, MT, BM, and FT). No attempts were made to contact authors or institutions to find additional sources. Any published manuscript presenting findings related to the association between egg consumption and bone mass density, English publications, and full texts available for extracting all relevant information were considered for study inclusion. The review included letters, reviews, observational studies, and experimental studies, while the exclusion criteria were books and grey literature publications and publications not in English.

### Data charting

Information on the paper identifiers (title, author, link), the country, year of publication, the study aims, study design, study location, the quantity of egg consumption, frequency of egg consumption, reported impact and effect size were extracted from the publications included in this review. The extracted information from each publication was compiled and summarized in Table 1.

## Results

The initial search using the predefined search terms from five databases yielded 315 studies. From the studies, 27 duplicates were removed, and the remaining 288 studies were screened. After screening, 283 studies were excluded because they did not meet the eligibility criteria. Five articles were sought for retrieval, of which three were excluded because two were studies on the impact of eggshell powder on bone mass among older people, and the last was a study conducted among children (Fig 1). Only two of the studies were included in the review. Table 1 shows the details of the included articles addressing the association between habitual egg consumption and bone mass density among older people.

### Overview of studies

Table 2 shows the overview of the two included studies [20,21]. The study conducted by Pujia et al. (2022) [20] delves into the association between egg consumption and bone health, particularly whole-body bone mineral density (BMD) and the T-score, a key metric for diagnosing osteoporosis and assessing treatment effectiveness. In an aging population of 176 individuals aged 65 years and older, the researchers aimed to unravel the potential impact of egg

**Table 1. Characteristics of included studies.**

| S/N | Title | Author (year of publication) | Study Aim | Publication Type | Gender distribution | Study Participants |
|---|---|---|---|---|---|---|
| 1 | Relationship between osteoporosis, multiple fractures, and egg intake in healthy elderly | Pujia et al. (2021) [20] | The aim of this research is to explore and understand the potential correlation between the intake of eggs and the density of bones among elderly individuals. This investigation seeks to delve into how egg consumption might impact bone density within this specific demographic. | Cross-sectional study | The number of females included in the study was 112, while the number of males was 64. | The participants were recruited between February 2013 and August 2016, and it was mentioned that this study constituted a secondary analysis of baseline data acquired from a study titled "Effect of the Mediterranean Diet on cognitive function in the elderly. |
| 2 | The association of red meat, poultry, and egg consumption with risk of hip fractures in elderly Chinese: A case–control study | Zeng et al., (2013) [21] | In order to investigate the potential connection between dietary consumption of various types of red meat, poultry (with or without skin), and eggs, and the likelihood of experiencing a hip fracture later on, a meticulously designed 1:1 matched case-control study was conducted. This study involved 646 pairs of elderly Chinese individuals hailing from Guangdong Province. The aim was to discern any associations between the mentioned dietary factors and the risk of hip fracture among this specific demographic group. | Case-Control study | The study comprised a total of 646 case patients and 646 matched control subjects, of which there were 484 female pairs and 162 male pairs. | According to the report, the cases consisted of hip fracture patients who were consecutively admitted to four hospitals: the First Affiliated Hospital of Sun Yat-sen University, Guangzhou Orthopedics Trauma Hospital, Guangdong General Hospital, and the Orthopedics Hospital of Baishi District in Jiangmen City, Guangdong province. It was mentioned that between June 2009 and January 2013, a total of 1281 patients aged 55 to 80, who had resided in Guangdong province for over 10 years, were admitted with hip fractures diagnosed within 2 weeks of their potential enrollment into the study and confirmed by X-ray image. |
| | | | | | | "The controls were individually matched to the cases based on sex and age. The inclusion and exclusion criteria for controls were the same as those for cases, except for a history of fracture. We had two sets of controls: 183 (28.3%) hospital controls who were in-patients admitted to the specified hospitals and Zhongshan Ophthalmic Center within one week and could be matched to the cases, and 463 (71.7%) community controls who were apparently healthy residents recruited from the same cities." |

consumption on bone density. Eggs were singled out due to their intriguing bioactive compounds, which might have positive implications for BMD. The study unearthed a statistically significant positive association between whole-body T-scores and egg consumption. Those who consumed more eggs exhibited higher T-scores, suggesting better bone density. Gender and body mass index (BMI) also influence bone health. Females had notably higher T-scores, and individuals with higher BMI tended to have better bone density. Intriguingly, multiple fractures were negatively associated with daily egg intake, implying that those who consumed more eggs were less prone to experiencing multiple fractures. HDL-C levels were linked positively with multiple fractures, indicating a potential role for cholesterol in bone health. This

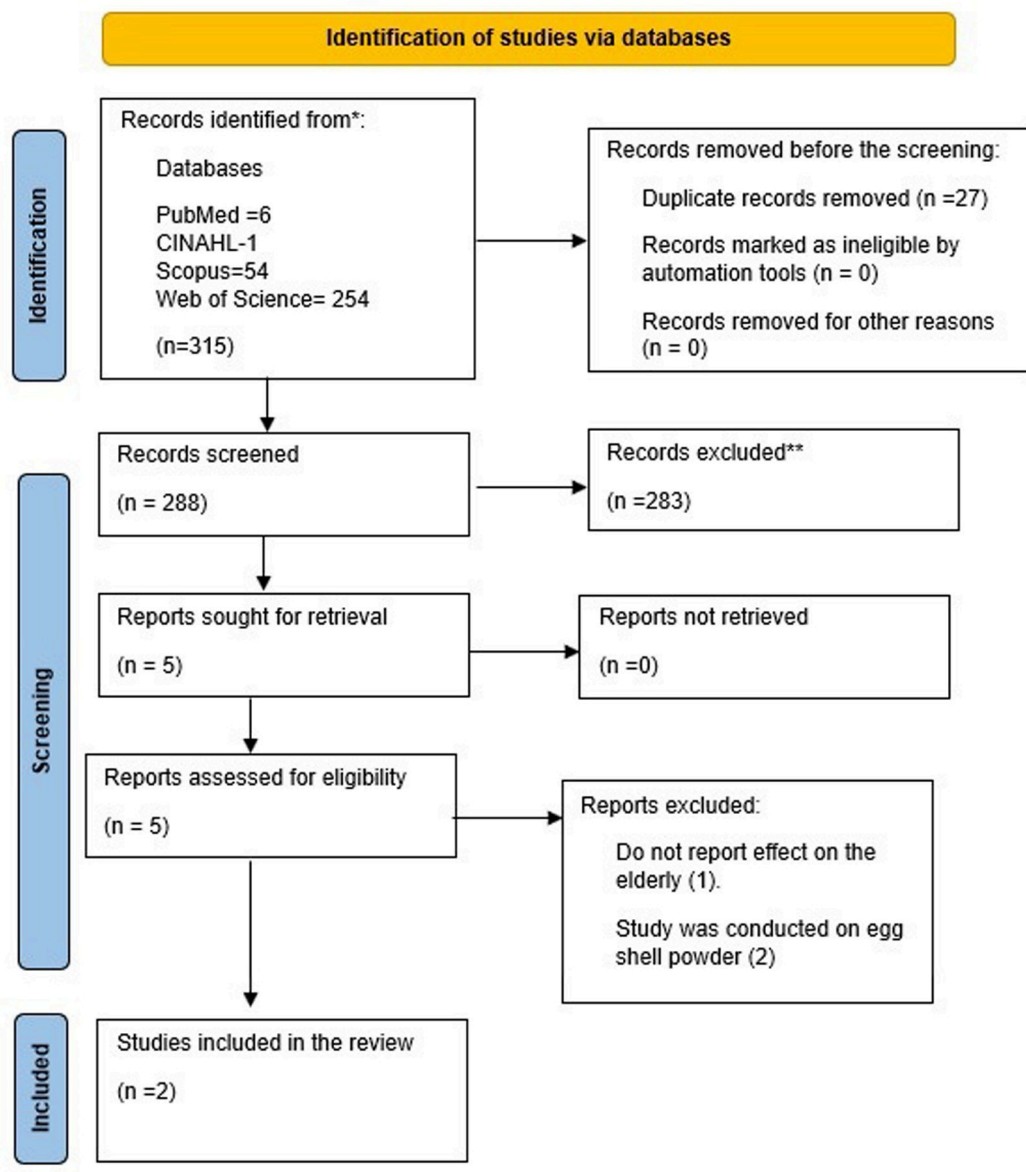

**Identification of studies via databases**

Records identified from*:

Databases

PubMed =6
CINAHL-1
Scopus=54
Web of Science= 254

(n=315)

Records removed before the screening:

Duplicate records removed (n =27)

Records marked as ineligible by automation tools (n = 0)

Records removed for other reasons (n = 0)

Records screened (n = 288)

Records excluded** (n =283)

Reports sought for retrieval (n = 5)

Reports not retrieved (n =0)

Reports assessed for eligibility (n = 5)

Reports excluded:

Do not report effect on the elderly (1).

Study was conducted on egg shell powder (2)

Studies included in the review (n =2)

*From:* Page MJ, McKenzie JE, Bossuyt PM, Boutron I, Hoffmann TC, Mulrow CD, et al. The PRISMA 2020 statement: an updated guideline for reporting systematic reviews. BMJ 2021;372:n71. doi: 10.1136/bmj.n71

**Fig 1. Study flow chart.**

study provides novel insights into the relationship between egg consumption and bone health in older people. It suggests that whole eggs might positively impact bone density, potentially reducing the risk of osteoporosis and fractures in older individuals. However, the study's cross-sectional nature limits its ability to establish causation and further research, ideally in the form of randomized controlled trials, is needed to confirm these findings. If substantiated, this

**Table 2. Study findings.**

| S/N | Title | Author (Year of publication) | Reported Quantity of Egg Consumed | Frequen cy of Egg Consumption | Reported effect on BMD | Conclusion |
|---|---|---|---|---|---|---|
| 1 | Relationship between osteoporosis, multiple fractures, and egg intake in healthy elderly | Pujia et al. (2021) [20] | Approximately 56% of the participants in the study reported consuming between 1 to 5 eggs per week, with an average weekly intake of one egg. | Weekly | The WB T-score exhibited a positive correlation with daily egg consumption (r = 0.16; P = 0.027). Conversely, multiple fractures demonstrated a negative correlation with egg intake (r = -0.39; P = 0.014). In regression analysis, the WB T-score continued to show a positive association with egg intake (B = 0.02; P = 0.02). | This cross-sectional study presents evidence supporting a positive association between whole egg consumption and whole-body bone mineral density (WB-BMD). If future randomized controlled trials confirm these findings, incorporating whole eggs into the diets of older adults could have a substantial public health impact, including potential benefits for osteoporosis prevention and reduced fracture risk. |
| 2 | The association of red meat, poultry, and egg consumption with risk of hip fractures in elderly Chinese: A case–control study | Zeng et al., (2013) [21] | The average daily intake for men was 20.2 grams, while for women it was 22.6 grams per day. | Daily | There was no observed evidence indicating a correlation between egg consumption and the risk of hip fracture (OR = 0.99; 95% CI: 0.63, 1.56). | This cross-sectional study presents evidence supporting a positive association between whole egg consumption and whole-body bone mineral density (WB-BMD). If future randomized controlled trials confirm these findings, incorporating whole eggs into the diets of older adults could have a substantial public health impact, including potential benefits for osteoporosis prevention and reduced fracture risk. |

study could pave the way for whole eggs as a viable dietary strategy for maintaining bone health in older people, addressing a crucial aspect of overall well-being in aging populations.

The research conducted by Zeng et al. (2013) [21] focused on exploring the potential link between the consumption of red meat, poultry (with or without skin), and eggs and the subsequent risk of hip fractures among older adult Chinese individuals. The study adopted a case–control design, with meticulous matching resulting in 646 pairs of participants from Guangdong Province, a coastal region in China. The study specifically targeted hip fractures, a significant concern among older people due to their impact on mobility and overall well-being. By investigating the dietary habits of these individuals, the researchers sought to identify any associations between food intake and hip fracture risk. The study's results indicated that, on average, men consumed around 20.2 g/d of the examined foods, while women's intake was slightly higher at 22.6 g/d. Upon analysis, the odds ratio (OR) for the risk of hip fracture about red meat, poultry, and eggs consumption was calculated at 0.99 (with a confidence interval of 0.63 to 1.56). This result suggests no conclusive evidence of a substantial association between egg consumption and the subsequent risk of hip fractures among the older adults' participants. While the study's findings do not indicate an egg consumption and hip fracture risk link, it is essential to consider the complexity of dietary patterns and potential confounding factors that may influence such outcomes. Overall, the research contributes to the ongoing discourse on nutrition and its impact on bone health, particularly in the context of the older adult population.

## Discussion

The scoping review highlights that only two studies show the relationship between habitual egg consumption and elderlies' bone fracture risk. Both studies had contradictory findings, though they used different study designs and had different study outcomes. This scoping review reveals a significant gap in the current literature regarding the potential correlation between regular egg consumption and bone mass density in the older adult population. The limited research highlights the necessity for additional investigation to gain a more comprehensive understanding of the potential causal relationship between these variables.

The limited number of primary studies investigating the relationship between egg consumption and bone health in older adults highlights a gap in research. This gap may be attributed to concerns about the cholesterol content in eggs and its potential association with an elevated risk of cardiovascular diseases. Notably, studies in Italy, Spain, and China have linked high egg consumption to an increased risk of overall mortality and mortality due to cancer [22–24]. Yet the protein content of eggs, such as ovotransferrin, has bone-preserving proprieties through inhibition of the bone resorption process [25]. There is also suggestive evidence that carotenoids in egg yolk could prevent bone loss [26,27]. Egg consumption is also associated with dyslipidemia, a condition characterized by abnormal blood lipid levels. Eggs, a rich source of dietary cholesterol, have historically been of concern because of their impact on blood cholesterol levels. The relationship between egg consumption and dyslipidemia can vary among individuals and is influenced by genetics and overall diet [28,29].

As individuals age, the concern for bone health becomes more pronounced, given the heightened vulnerability to bone fragility. This increased susceptibility poses a significant risk of falls and fractures, with potentially severe consequences, particularly for the elderly population [17,30]. Recognizing this vulnerability, it becomes imperative to provide older people with comprehensive lifestyle and dietary recommendations that empower them to make informed choices to maintain their health and well-being [31]. Interestingly, emerging evidence suggests that eggs, often overlooked due to concerns about their cholesterol content, contain compounds that could contribute to bone development and enhance bone strength. Recognizing eggs' potential role in bone health signifies the importance of considering them as a dietary component for the aging population. While the potential cardiovascular risks of consuming cholesterol-rich foods like eggs have garnered significant attention, the flip side of the coin reveals a more nuanced story. Recent research has highlighted that some cholesterol in eggs may possess.

Protective qualities against dyslipidemia, a condition characterized by abnormal levels of lipids in the blood, are intricately linked to cardiovascular health [32,33]. These contradictory findings emphasize the complexity of the relationship between egg consumption, cholesterol, and overall health, suggesting that a blanket assessment of eggs as detrimental may only partially be accurate [34].

In the broader context of bone health, eggs emerge as a potential source of nutrients that could contribute to maintaining bone density and reducing the risk of fractures among older people [35]. Being abundant in protein, eggs supply vital amino acids crucial for forming and maintaining the bone matrix [2]. Moreover, eggs serve as a natural reservoir of vitamin D, a crucial nutrient facilitating calcium absorption and bone mineralization. This distinctive nutritional profile positions eggs as a singular dietary choice that fulfills the nutritional needs for promoting bone health in aging individuals [36].

Addressing the current research gap concerning the relationship between egg consumption and bone mass density is essential for advancing our knowledge of bone health and advocating evidence-based dietary recommendations for older people. As the population ages, preserving

mobility and reducing the risk of falls and fractures becomes increasingly paramount [31]. Therefore, adopting a comprehensive approach that includes dietary components like eggs, appropriate exercise, and other lifestyle adjustments could play a pivotal role in promoting healthier aging.

One of the strengths of this paper was assessing the effect of habitual egg consumption on bone mass density in a larger population than reported in a single study, which helps estimate effect size better and makes it easier to make recommendations to the broader population of older adults scientifically. One major limitation of the study is the absence of studies on the subject of discourse, which would provide an excellent way to dispel myths and fads surrounding the habitual consumption of eggs with sufficient scientific evidence on the subject across a wider population. Also, the study is limited in the uniformity of study design across the studies included.

In conclusion, the limited research exploring the potential link between regular egg consumption and bone mass density, especially among older people, highlights the necessity for more extensive investigation. The reluctance to delve into this relationship may partly stem from concerns regarding cholesterol content, which has historically overshadowed the potential health benefits of eggs. Nonetheless, considering the older adults population's susceptibility to bone-related issues and the urgency of taking proactive measures to prevent falls and fractures, it becomes imperative to examine the potential contributions of eggs to bone health. As research advances, a balanced assessment of how eggs impact bone and cardiovascular health should guide dietary recommendations. This approach acknowledges the complexity of the nutritional composition of eggs and their potential role in enhancing the well-being of the aging population.

## Conclusion

The existing literature on the relationship between habitual egg consumption and bone mass density among older individuals reveals a significant research gap, with only a limited number of studies exploring this crucial association. The hesitancy to delve into this relationship may stem from historical concerns about cholesterol content, overshadowing the potential health benefits of eggs. However, considering the heightened vulnerability of older adults to bone-related issues and the imperative of proactive measures to prevent falls and fractures, a comprehensive examination of the potential contributions of eggs to bone health is essential. As research advances, a balanced assessment considering the complexity of the nutritional composition of eggs should guide dietary recommendations for the aging population, acknowledging both potential benefits and risks.

## Recommendation

Future research endeavors should focus on bridging the current gap in understanding the relationship between habitual egg consumption and bone health among older individuals. Given the conflicting reports and limited studies on this subject, a more extensive investigation with diverse study populations and standardized study designs is essential. This will contribute to dispelling myths and fads surrounding egg consumption and provide robust scientific evidence to inform dietary recommendations for the aging population. In addition to promoting bone health, further research can explore the broader implications of egg consumption on cardiovascular health in older individuals, ensuring a holistic understanding of the potential benefits and risks associated with incorporating eggs into the diet of the older adults.

## Supporting information

**S1 Checklist. Preferred Reporting Items for Systematic reviews and Meta-Analyses extension for Scoping Reviews (PRISMA-ScR) checklist.**
(DOCX)

**S1 Appendix. Search strategy and strings.**
(DOCX)

## Acknowledgments

We extend our heartfelt gratitude to Prof. Morenike Ukpong for her invaluable contribution in reviewing the manuscript. Her keen insights and rigorous evaluation significantly enhanced the quality and rigor of this work. Her expertise and constructive feedback were indispensable in refining our manuscript and advancing the discourse in this field.

## Author Contributions

**Conceptualization:** Mobolaji T. Olagunju, Olunike R. Abodunrin, Ifeoluwa O. Omotoso, Ifeoluwa E. Adewole, Oluwabukola M. Ola, Chukwuemeka Abel, Folahanmi T. Akinsolu.

**Data curation:** Mobolaji T. Olagunju, Olunike R. Abodunrin, Ifeoluwa O. Omotoso, Ifeoluwa E. Adewole, Oluwabukola M. Ola, Chukwuemeka Abel, Folahanmi T. Akinsolu.

**Formal analysis:** Mobolaji T. Olagunju, Olunike R. Abodunrin, Ifeoluwa O. Omotoso, Ifeoluwa E. Adewole, Oluwabukola M. Ola, Chukwuemeka Abel, Folahanmi T. Akinsolu.

**Investigation:** Mobolaji T. Olagunju, Olunike R. Abodunrin, Ifeoluwa O. Omotoso, Ifeoluwa E. Adewole, Oluwabukola M. Ola, Chukwuemeka Abel.

**Methodology:** Mobolaji T. Olagunju, Olunike R. Abodunrin, Ifeoluwa O. Omotoso, Ifeoluwa E. Adewole, Oluwabukola M. Ola, Chukwuemeka Abel, Folahanmi T. Akinsolu.

**Project administration:** Mobolaji T. Olagunju, Oluwabukola M. Ola, Chukwuemeka Abel.

**Writing – original draft:** Mobolaji T. Olagunju, Olunike R. Abodunrin, Ifeoluwa O. Omotoso, Ifeoluwa E. Adewole, Oluwabukola M. Ola, Chukwuemeka Abel, Folahanmi T. Akinsolu.

**Writing – review & editing:** Mobolaji T. Olagunju, Olunike R. Abodunrin, Ifeoluwa O. Omotoso, Ifeoluwa E. Adewole, Oluwabukola M. Ola, Chukwuemeka Abel, Folahanmi T. Akinsolu.

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
