## [Decision Letter · Decision Letter 0]

20 Dec 2023

PGPH-D-23-01926

Egg Consumption and Bone Mass Density among the Elderly: A Scoping Review

Dear Dr. Akinsolu,

Thank you for submitting your manuscript to PLOS Global Public Health. After careful consideration, we feel that it has merit but does not fully meet PLOS Global Public Health’s publication criteria as it currently stands. Therefore, we invite you to submit a revised version of the manuscript that addresses the points raised during the review process.

The comments from our expert reviewers are very critical and you are entreated to respond to each of it adequately and revise the manuscript accordingly. Please, any changes to the original manuscript MUST be clearly HIGHLIGHTED in Yellow. Refusal to do this will either delay the progress of your manuscript or reject it.

We look forward to receiving your revised manuscript.

Kind regards,

Professor Razak M Gyasi, PhD, PD

Academic Editor

Journal Requirements:

1. Please provide separate figure files in .tif or .eps format only and remove any figures embedded in your manuscript file. Please also ensure all files are under our size limit of 10MB.

Additional Editor Comments (if provided):

Reviewers' comments:

Reviewer's Responses to Questions

**Comments to the Author**

1. Does this manuscript meet PLOS Global Public Health’s publication criteria? Is the manuscript technically sound, and do the data support the conclusions? The manuscript must describe methodologically and ethically rigorous research with conclusions that are appropriately drawn based on the data presented.

Reviewer #1: Yes

Reviewer #2: Partly

Reviewer #3: Partly

Reviewer #4: Yes

2. Has the statistical analysis been performed appropriately and rigorously?

Reviewer #1: Yes

Reviewer #2: Yes

Reviewer #3: No

Reviewer #4: Yes

3. Have the authors made all data underlying the findings in their manuscript fully available (please refer to the Data Availability Statement at the start of the manuscript PDF file)?

Reviewer #1: Yes

Reviewer #2: Yes

Reviewer #3: No

Reviewer #4: No

4. Is the manuscript presented in an intelligible fashion and written in standard English?

Reviewer #1: Yes

Reviewer #2: Yes

Reviewer #3: Yes

Reviewer #4: Yes

5. Review Comments to the Author

Reviewer #1: This is an interesting paper that is somewhat speculative. Most of its value is in offering a hypothesis that egg consumption should be more closely examined as protective of bone health, based on what we know about egg nutritional composition.

Reviewing 280+ research studies and then finding that only two directly investigate connections between bone health and egg consumption, is a bit strange, especially since as far as I can tell, the Guangdong study did not find a clear relationship (did it separately examine effects of red meat consumption versus egg consumption?). The other study was not a controlled study and causation is difficult to isolate. So all that the literature review really tells us is that no high quality study of the role of egg consumption in protecting bone health has been done. I hope that the authors are themselves working on a study to investigate this question, and would rather support publication of this data, followed by their contribution to answering this question with carefully designed research. However, I can also appreciate the contribution that a literature review makes.

At the very end of the paper, there is a paragraph on p. 11 that seems overly congratulatory.

--""One of the strengths of this paper was assessing the effect of habitual bone consumption on bone mass density in a larger population than reported in a single study, which helps estimate effect size better and makes it easier to make recommendations to the broader population of older adults scientifically." :

First, there is a major typo here: "habitual BONE consumption"!?

Second, actually, I don't find that you have effectively assessed the effect of egg consumption. I don't think either study is valid enough to really estimate effect size. Also, two papers is admittedly more than one, but it's not much larger - so don't overexaggerate the significance here.

"One major limitation of the study is the dearth of studies on the subject of discourse,

which would provide an excellent way to dispel myths and fads surrounding the habitual

consumption of eggs with sufficient scientific evidence on the subject across a wider population.

Also, the study is limited in the uniformity of study design across the studies included."

--It appears that your goal is to dispel myths and fads, which seems a bit inappropriate given the limited evidence. I would add that the study is limited in that clear causality is not demonstrated in either study, and moreover, that a thorough research on the topic has yet to be carried out.

All in all, an interesting study that is yet very limited in its ability to draw conclusions, but could lay helpful groundwork for future research.

Reviewer #2: Some what the articles were clear, correct, and unambiguous. Some grammatical errors were found along with significant plagiarism. Although plagiarism was found, I think it below 25%. So, the author should be able to paraphrase to solve existing paraphrases.

Reviewer #3: The manuscript titled "Egg Consumption and Bone Mass Density among Older Adults: A Scoping Review" provides a comprehensive exploration of the relationship between egg consumption and bone health in the aging population. Several critical points have been identified for improvement:

Language Refinement:

To ensure appropriateness in language, the term "elferly" throughout the manuscript should be replaced with "older adults." This adjustment contributes to the professionalism and clarity of the text.

Structural Enhancement:

The summary currently emphasizes the information filtering process, which is more aligned with the methodology. To enhance clarity, it is recommended to shift the primary focus of the summary to the key findings discovered during the review. This modification ensures that the most significant outcomes are prominently featured in the results section.

Reference Completion:

Line 71 indicates an incomplete reference. To maintain the scholarly integrity of the manuscript, it is essential to provide the necessary information and complete the reference accurately.

Avoiding Plagiarism:

Several instances throughout the manuscript involve verbatim use of text from the original references, which poses a risk of plagiarism. To address this concern, it is imperative to rephrase and paraphrase these sections, ensuring that the content is in the author's own words while appropriately citing the sources.

Registration Justification:

Even though the manuscript is a review, it is crucial to provide a justification for the absence of a registration. Explaining the rationale behind this decision will enhance transparency and help address any concerns related to research accountability.

Addressing these points will contribute to the overall quality, clarity, and integrity of the manuscript, ensuring it meets the highest standards of academic writing.

Reviewer #4: A good focus on the critical nutrient source (eggs) and effects on bones especially in older persons. the methods were good despite not getting enough papers for review. indicate that interpretation should be with caution.

following this review, the authors might consider a systematic review on the same topic to see of there is a wider scope of publications on the same topic, the authors have suggested the need for more studies, however, a systematic and or meta-analysis might also be useful

6. PLOS authors have the option to publish the peer review history of their article (what does this mean?). If published, this will include your full peer review and any attached files.

**Do you want your identity to be public for this peer review?** For information about this choice, including consent withdrawal, please see our Privacy Policy.

Reviewer #1: No

Reviewer #2: **Yes: **Ok, I will not do that.

Reviewer #3: No

Reviewer #4: No

---

## [Decision Letter · Decision Letter 1]

5 Mar 2024

PGPH-D-23-01926R1

Egg Consumption and Bone Mass Density among the Elderly: A Scoping Review

Dear Dr. Akinsolu,

Thank you for submitting your manuscript to PLOS Global Public Health. After careful consideration, we feel that it has merit but does not fully meet PLOS Global Public Health’s publication criteria as it currently stands. Therefore, we invite you to submit a revised version of the manuscript that addresses the points raised during the review process.

Our reviewer who is an expert in the field has raised very critical concerns regarding the similarity index of the paper which has also been noted by the editorial team. Please, work on the draft to make sure these issues and other pointed out are fully addressed. 

We look forward to receiving your revised manuscript.

Kind regards,

Prof Razak M Gyasi, PhD, PD

Academic Editor

Journal Requirements:

Additional Editor Comments (if provided):

Reviewers' comments:

Reviewer's Responses to Questions

**Comments to the Author**

1. If the authors have adequately addressed your comments raised in a previous round of review and you feel that this manuscript is now acceptable for publication, you may indicate that here to bypass the “Comments to the Author” section, enter your conflict of interest statement in the “Confidential to Editor” section, and submit your "Accept" recommendation.

Reviewer #3: (No Response)

Reviewer #4: All comments have been addressed

2. Does this manuscript meet PLOS Global Public Health’s publication criteria? Is the manuscript technically sound, and do the data support the conclusions? The manuscript must describe methodologically and ethically rigorous research with conclusions that are appropriately drawn based on the data presented.

Reviewer #3: Partly

Reviewer #4: Yes

3. Has the statistical analysis been performed appropriately and rigorously?

Reviewer #3: Yes

Reviewer #4: Yes

4. Have the authors made all data underlying the findings in their manuscript fully available (please refer to the Data Availability Statement at the start of the manuscript PDF file)?

Reviewer #3: Yes

Reviewer #4: Yes

5. Is the manuscript presented in an intelligible fashion and written in standard English?

Reviewer #3: Yes

Reviewer #4: Yes

6. Review Comments to the Author

Reviewer #3: Thank you for making the modifications to the manuscript. Its quality has improved. However, I believe some pertinent modifications are still required.

Firstly, regarding the similarity analysis; excluding the published preprint, it is important to focus on the information presented in tables 1 and 2, as the objectives, conclusions, and other data are exactly like those in the cited article. I recommend condensing the information and avoiding the use of the same text as in the cited documents.

Secondly, concerning the abstract, the results section can focus solely on the findings of this article. There is no need for a summary of the method; instead, the essence of the analysis findings is required.

I appreciate your attention to these matters.

Reviewer #4: (No Response)

7. PLOS authors have the option to publish the peer review history of their article (what does this mean?). If published, this will include your full peer review and any attached files.

**Do you want your identity to be public for this peer review?** For information about this choice, including consent withdrawal, please see our Privacy Policy.

Reviewer #3: No

Reviewer #4: No

---

## [Decision Letter · Decision Letter 2]

10 Apr 2024

Egg Consumption and Bone Mass Density among the Elderly: A Scoping Review

PGPH-D-23-01926R2

Dear Dr. Akinsolu,

We are pleased to inform you that your manuscript 'Egg Consumption and Bone Mass Density among the Elderly: A Scoping Review' has been provisionally accepted for publication in PLOS Global Public Health.

Best regards,

Razak M Gyasi, PhD, PD

Academic Editor

Reviewer Comments (if any, and for reference):

Reviewer's Responses to Questions

**Comments to the Author**

1. If the authors have adequately addressed your comments raised in a previous round of review and you feel that this manuscript is now acceptable for publication, you may indicate that here to bypass the “Comments to the Author” section, enter your conflict of interest statement in the “Confidential to Editor” section, and submit your "Accept" recommendation.

Reviewer #3: All comments have been addressed

2. Does this manuscript meet PLOS Global Public Health’s publication criteria? Is the manuscript technically sound, and do the data support the conclusions? The manuscript must describe methodologically and ethically rigorous research with conclusions that are appropriately drawn based on the data presented.

Reviewer #3: Yes

3. Has the statistical analysis been performed appropriately and rigorously?

Reviewer #3: Yes

4. Have the authors made all data underlying the findings in their manuscript fully available (please refer to the Data Availability Statement at the start of the manuscript PDF file)?

Reviewer #3: Yes

5. Is the manuscript presented in an intelligible fashion and written in standard English?

Reviewer #3: Yes

6. Review Comments to the Author

Reviewer #3: Thank you very much for addressing all the recommendations made to the manuscript. I consider that you have fulfilled the requested guidelines. Congratulations

7. PLOS authors have the option to publish the peer review history of their article (what does this mean?). If published, this will include your full peer review and any attached files.

**Do you want your identity to be public for this peer review?** For information about this choice, including consent withdrawal, please see our Privacy Policy.

Reviewer #3: No
